# Novel 1,3,4-Thiadiazole Derivatives: Synthesis, Antiviral Bioassay and Regulation the Photosynthetic Pathway of Tobacco against TMV Infection

**DOI:** 10.3390/ijms24108881

**Published:** 2023-05-17

**Authors:** Huanlin Zheng, Fanglin Wen, Chengzhi Zhang, Rui Luo, Zhibing Wu

**Affiliations:** 1National Key Laboratory of Green Pesticide, Key Laboratory of Green Pesticide and Agricultural Bioengineering, Ministry of Education, Center for R&D of Fine Chemicals of Guizhou University, Guiyang 550025, China; 2College of Life Sciences, Guizhou University, Guiyang 550025, China

**Keywords:** 1,3,4-thiadiazole, antiviral activity, tobacco mosaic virus (TMV), paraffin section, photosynthesis, chlorophyll, peroxidation

## Abstract

Tobacco mosaic virus (TMV) is a systemic virus that poses a serious threat to crops worldwide. In the present study, a series of novel 1-phenyl-4-(1,3,4-thiadiazole-5-thioether)-1*H*-pyrazole-5-amine derivatives was designed and synthesized. In vivo antiviral bioassay results indicated that some of these compounds exhibited excellent protective activity against TMV. Among the compounds, **E_2_** (EC_50_ = 203.5 μg/mL) was superior to the commercial agent ningnanmycin (EC_50_ = 261.4 μg/mL). Observation of tobacco leaves infected with TMV-GFP revealed that **E_2_** could effectively inhibit the spread of TMV in the host. Further plant tissue morphological observation indicated that **E_2_** could induce the tight arrangement and alignment of the spongy mesophyll and palisade cells while causing stomatal closure to form a defensive barrier to prevent viral infection in the leaves. In addition, the chlorophyll content of tobacco leaves was significantly increased after treatment with **E_2_**, and the net photosynthesis (*Pn*) value was also increased, which demonstrated that the active compound could improve the photosynthetic efficiency of TMV-infected tobacco leaves by maintaining stable chlorophyll content in the leaves, thereby protecting host plants from viral infection. The results of MDA and H_2_O_2_ content determination revealed that **E_2_** could effectively reduce the content of peroxides in the infected plants, reducing the damage to the plants caused by oxidation. This work provides an important support for the research and development of antiviral agents in crop protection.

## 1. Introduction

Plant virus diseases seriously threaten the development of global agricultural economies, crop yields and quality. Tobacco mosaic virus (TMV), an RNA virus, has a wide number of hosts, including over 885 plant species in 65 families. It causes mosaicism, poor plant growth and deformed leaf symptoms, resulting in severe economic losses in agriculture. TMV is a systemic plant viral pathogen, and there are no effective control methods or treatments available to protect plants against TMV infection. At present, some commercial agents, such as ningnanmycin and ribavirin, exhibit unsatisfactory antiviral activities [1,2,3,4]. Therefore, it is still necessary to find efficient antiviral agents.

Studies of the effect of viral infection on the physiological performance and productivity parameters in plants have shown that the symptoms of infected plants usually include mottling, mosaic, chlorosis and yellowing. In addition, the photosynthetic capacity of the plant is therefore reduced [5,6,7]. Photosynthesis is a ubiquitous physiological process in plants that is the material basis and energy source for plant growth and physiological metabolism, and the strength of photosynthesis affects the physiological activity and yield of plants. Nevertheless, photosynthetic activity is determined by the chlorophyll content, which also reflects the health of plants and has an important impact on plant growth [8,9,10]. The chlorophyll content and the net photosynthesis (*Pn*) decrease in the leaves of infected plants [11,12].

In previous studies, 1,3,4-thiadiazole derivatives with broad activities, such as antifungal [13], antiviral [14], insecticidal [15], anticancer [16] and antibacterial activities [17], have been widely used for molecular design in pharmaceuticals and pesticides. Recently, 1,3,4-thiadiazole derivatives were revealed to have potential anti-TMV activity, and their mechanism of action has not been reported [14,18]. In the present work, combining the functional group pyrazole and 1,3,4-thiadiazole, a series of novel 1-phenyl-4-(1,3,4-thiadiazole-5-thioether)-1*H*-pyrazole-5-amine derivatives was designed and synthesized (Figure 1). In vivo antiviral activity evaluation revealed that some of these compounds exhibited excellent protective activity against TMV that was better than the commercially agent ningnanmycin. Plant tissue morphological observations and physiological and biochemical determinations were used for mechanism studies, and it was found that this series of compounds could not only induce leaf tissue development to enhance photosynthetic activity and induce stomatal closure to defend against virus invasion but could also effectively reduce the content of peroxides in the infected plants, reducing the damage to the plants caused by oxidation.

## 2. Results

### 2.1. Chemistry

As shown in Figure 1, intermediate **B** was synthesized by a cyclization reaction of raw material **A** and phenylhydrazine, followed by hydrazinolysis with hydrazine hydrate to obtain intermediate **C**, which was first reacted with CS_2_ and KOH in the presence of ethanol and then cyclized with concentrated sulfuric acid to obtain intermediate **D**. The title compounds **E_1_**–**E_28_** were obtained by the nucleophilic substitution reaction of **D** and different benzyl groups. All the target compounds were confirmed by ^1^H NMR, ^13^C NMR, ^19^F NMR and HRMS, and detailed physical and spectral data are provided in the Appendix A.

Moreover, the crystal structure of compound **E_3_** (Figure 2) was determined by single-crystal X-ray diffraction analysis. The skeleton of **E_3_** consisted of a pyrazole ring and a 1,3,4-thiadiazole ring, which were connected by C (9) and C (10). The benzene ring and NH_2_ group were connected to the pyrazole ring at the 1-position and 5-position by N (13)—C (4) and N (5)—C (12), respectively. The benzyl group was connected with the 1,3,4-thiadizole ring by C (8)—S (1)—C (7). The X-ray crystal structure data of **E_3_** can be downloaded from the Cambridge Crystallographic Data Centre (CCDC 2182744), and detailed crystallographic data are provided in the Appendix A.

### 2.2. Antiviral Activity

Preliminary in vivo antiviral activity results for the title compounds at a concentration of 500 μg/mL are shown in Table 1. Among them, compound **E_8_** (59.2%) exhibited curative activity comparable to that of ningnanmycin (58.0%), and compounds **E_2_** (65.1%) and **E_17_** (61.3%) displayed better protective activity than that of ningnanmycin (54.6%). In addition, compound **E** also exhibited certain inactivating activity (Table 2). EC_50_ value testing also showed that **E_2_** (203.5 μg/mL) exhibited excellent protective activity that was better than ningnanmycin (261.4 μg/mL) (Table 3), and detailed inhibition rate and images of title compounds against TMV are provided in the Appendix A. As shown in Figure 3, visual symptoms of rolling and mottling were observed on the upper leaves of the S group during TMV infection.

### 2.3. Effect of E_2_ against TMV Spread in N. benthamiana

The protective activity of **E_2_** was further verified by observing the infection process of TMV with a GFP tag under UV light (365 nm). As shown in Figure 4, in the S groups, green fluorescence was first observed at 4 dpi (Figure 4A) in inoculated leaves of *N. benthamiana*, was transmitted to the upper leaves at 8 dpi (Figure 4B) and became increasingly stronger on the top leaves at 10 dpi (Figure 4C). Conversely, in the TS groups, the intensity of green fluorescence was much less than that of the S groups at 10 dpi (Figure 4E), and almost no fluorescence was found at 8 dpi (Figure 4D).

### 2.4. Tissue Morphology Observed in Paraffin Sections

As shown in Figure 5, the transverse sections of *N. benthamiana* subjected to four different treatments were observed. Compared with the CK group (Figure 5A), the T group and TS group had epidermal cells that became larger, and the spongy mesophyll cells and palisade cells were closely aligned and had a larger distribution density (Figure 5B,D). However, morphological analysis demonstrated that the epidermal cells were dissociated and that the spongy mesophyll cells disappeared after infection with TMV in the S group (Figure 5C).

### 2.5. Photosynthesis and Gas Exchange

The statistical analysis shown in Figure 6A indicates that the *Pn* value in the CK, T and TS groups did not show apparent changes from 2 to 6 dpi but significantly declined in the S group. In the CK group, the *Pn* value peaked at 4 dpi (8.5 mmol/m^2^/s) and then decreased at 6 dpi (6.7 mmol/m^2^/s). In the T group, the *Pn* value reached a maximum value at 2 dpi (10.1 mmol/m^2^/s) and remained unchanged at 4 and 6 dpi (8.8 mmol/m^2^/s). In the TS group, it first decreased and then increased at 2, 4 and 6 dpi (6.9, 6.3 and 7.6 mmol/m^2^/s). In the S group, the *Pn* value at 2 and 4 dpi was 6.9 and 6.3 mmol/m^2^/s, respectively, while it dramatically dropped at 6 dpi (2.2 mmol/m^2^/s). Furthermore, the *Tr* and *gs* (Figure 6B,C) values were consistent with the trend of the *Pn* in each group. At 6 dpi, the *Tr* values in the CK, T, S and TS groups were 0.84, 1.36, 0.34 and 0.85 mmol H_2_O m^2^/s, respectively, and the *gs* values were 0.088, 0.140, 0.038 and 0.081 mol H_2_O m^2^/s, respectively. The *Ci* value (Figure 6D) decreased continuously in the CK, T and TS groups but first decreased and then increased in the S group from 2 to 6 dpi.

### 2.6. Stomatal Analysis of Tobacco Leaves

Analysis of the stomatal morphology and measurement statistics for the leaves in the four groups demonstrated that both the stomatal area (Figure 7E) and stomatal aperture (Figure 7F) were enlarged in the T group (1.17 μm^2^ and 0.33 μm, respectively) and S group (0.98 μm^2^ and 0.35 μm, respectively) compared to the CK group (0.82 μm^2^ and 0.26 μm, respectively) (Figure 7A). The increase was observed in the TS group (1.02 μm^2^ and 0.27 μm, respectively), which is consistent with the CK group. The detailed data of the average values of the stomatal area and stomatal aperture are shown in Table 4.

### 2.7. Chlorophyll, H_2_O_2_ and MDA Content Analysis in Tobacco Leaves

The contents of chlorophyll a (*C*_a_), chlorophyll b (*C*_b_) and total chlorophyll (*C*_t_) in the four treatment groups are shown in Figure 8. Compared with the S group, the contents of *C*_a_, *C*_b_ and *C*_t_ in the other three treatment groups were higher from 2 to 6 dpi and reached the highest value (0.57, 0.24 and 0.81 mg/g) at 4 dpi in the TS group, which was comparable to the T group (0.61, 0.24 and 0.84 mg/g) and higher than the CK group (0.52, 0.21 and 0.73 mg/g). The *C*_a_, *C*_b_ and *C*_t_ contents declined at 6 dpi in the TS group (0.36, 0.19 and 0.54 mg/g, respectively), which were lower than those in the CK group (0.46, 0.24 and 0.7 mg/g, respectively) and T group (0.57, 0.27 and 0.84 mg/g, respectively). Furthermore, the leaves of the four groups at 4 dpi under UV transillumination are shown in Figure 8D.

The MDA and H_2_O_2_ contents for the four different groups are displayed in Figure 9A,B. Among them, the MDA content slightly increased from 2 to 6 dpi in the two groups, which was 20.43–23.41 nmol/g in the CK group and 20.97–27.29 nmol/g in the TS group, with the S group always showing a higher content (26.12–41.15 nmol/g) and the T group always showing a lower content (19.71–21.86 nmol/g). Compared to the CK group, the H_2_O_2_ content remained low (4.40–3.67 μmol/g) in the S group, from 2 to 4 dpi, and significantly increased at 6 dpi (11.37 μmol/g). However, the H_2_O_2_ content was 4.4–6.2 μmol/g in the TS group, which was comparable to the CK group (5.6–6.6 μmol/g) from 2 to 6 dpi.

## 3. Discussion

In vivo antiviral activity results of title compounds **E_1_**–**E_28_** against TMV revealed that some exhibited excellent protective activities. Further experiments of **E_2_** in TMV-infected *N. benthamiana* as observed by TMV-GFP demonstrated that the active compound effectively inhibited the spread of TMV in the host by reducing rolled and mottling symptoms of infected leaves, which could be used for protecting the tobacco against viruses.

Leaf structure is an important determinant of leaf photosynthetic characteristics [19]. The spongy mesophyll and the palisade tissue of normal leaves contain abundant chlorophyll, which is distributed roughly equally between palisade tissue and spongy mesophyll cells [20,21]. However, the chlorophyll content in developed palisade tissue is higher than that in thicker palisade tissue leaves [22]. As shown in the morphology observations of paraffin sections (Figure 5), the spongy mesophyll cells were dissociated, and the density of the epidermal cells was increased in the S group; however, the developed epidermal cells and density of the spongy mesophyll cells were both observed in the T and TS groups that were first treated with title compound **E_2_**. The analysis of chlorophyll (*C*_a_, *C*_b_ and *C*_t_) content in four different groups showed that the chlorophyll content decreased with the progression of virus infection, which was consistent with Koiwa’s description [23]. Combining the two experimental results, we speculated that the active compound **E_2_** could induce the tight arrangement and alignment of spongy mesophyll and palisade cells, prompting them to contain more chlorophyll and making the host more resistant to TMV infection.

The *Pn* value is correlated with the chlorophyll content per unit leaf mass, and a lower chlorophyll content results in a lower *Pn* value [24,25]. Further photosynthetic parameter measurement results revealed that the *Pn*, *Tr* and *gs* values of tobacco leaves were reduced upon TMV infection, which indicated that the virus could change the chlorophyll content and affect the photosynthesis process [26,27,28,29]. However, after treatment with **E_2_**, the *Pn*, *Tr* and *gs* values were obviously increased and remained stable after TMV infection, revealing that active compounds could enhance photosynthesis efficiency in TMV-infected tobacco leaves by maintaining stable chlorophyll content. In addition, stomatal morphological analysis indicated that **E_2_** could induce stomatal closure to achieve a healthy state during TMV infection progress and prevent further infection with viruses through stomata [30], but there was no significant change in the *Pn* value affected by stomatal closure.

Reactive oxygen species (ROS), as important signal transduction molecules that are increased dramatically under stress conditions, play an important role in plant biological activities [31,32,33]. H_2_O_2_, as the main content of ROS, is one of the indicators of the toxic effects of a high level of oxidative stress [34,35,36]. Lipid peroxidation is a well-known index for determining the extent of oxidative stress, and MDA content is found to be highly correlated with oxidative damage in biotic and abiotic plants [37,38]. In the present work, both MDA and H_2_O_2_ contents were significantly increased in the S group and slightly increased in the TS group, which demonstrated that **E_2_** could effectively reduce the content of peroxides when plants are infected with TMV, reducing the damage to the plants caused by oxidation [34].

## 4. Materials and Methods

### 4.1. Instruments and Materials

The ^1^H nuclear magnetic resonance (NMR), ^13^C NMR and ^19^F NMR spectra were obtained by a JEOL-ECX 500 NMR spectrometer (JEOL Corporation, Tokyo, Japan) and Bruker 400 NMR spectrometer (Bruker Corporation, Karlsruhe, Germany) with tetramethylsilane (TMS) as an internal standard and CDCl_3_ or DMSO-*d*_6_ as the solvent. High-resolution mass spectrometry (HRMS) data were obtained on a Thermo Scientific Q Exactive (Thermo Scientific, Waltham, MA, USA). The X-ray crystallographic data were collected by a Bruker APEX-II CCD diffractometer (Bruker Corporation, Germany). A Cytation 5 Microplate Reader (Bio Tek Corporation, Winooski, VT, USA) was used to measure absorbance. The photosynthetic parameters were determined by a gas exchange system (HED-GH20 photosynthesis system, Horde Electronic Technology Co., Ltd., Shandong, China). The paraffin samples were sectioned on a Reichert Hisrostat Slicer (Reichert Hisrostat, New York, USA). The cell morphology and stomata were observed by an Olympus-CX33 microscope (Olympus Co., Ltd., Tokyo, Japan). The melting points of compounds were recorded on an XT-4 binocular microscope melting point apparatus (TEOH Corporation, Beijing, China) and were uncorrected. All reagents and solvents were purchased from commercial sources and used without further purification and drying. The key raw material ethyl (ethoxymethylene) cyanoacetate (**A**) was purchased from a commercial source (Adamas Reagent Co., Ltd., Shanghai, China).

### 4.2. Plants, TMV, and TMV-GFP

Three kinds of tobacco plants (*Nicotiana glutinosa*, *Nicotiana benthamiana* and *Nicotiana tabacum* cv. K326) were used for the experiments, and the plants were cultivated and maintained in growth chambers at a temperature of 28 °C, a day period with a 16 h photoperiod, a night cycle of 8 h, 25 °C and 85% humidity. *N. glutinosa* was used for the in vivo antiviral bioassay, *N. benthamiana* was used for mechanism studies, and *N. tabacum* cv. K326 was used for virus preservation. The tobacco mosaic virus was purchased from the Wuhan Institute of Virology, Chinese Academy of Sciences and preserved in *N. tabacum* cv. K326. *Agrobacterium tumefaciens* containing TMV-C58C1 (TMV-GFP) was provided by Shi, PhD, of Guizhou Medical University.

### 4.3. Synthetic Procedure [39,40,41]

#### 4.3.1. General Procedure for the Preparation of Intermediate **B**


Raw material **A** (2.0 mol) was dissolved in ethanol (30 mL), and then phenylhydrazine (2.3 mol) was added, followed by stirring at 95 °C for 3 h. The solvent was removed under vacuum, and the crude mixture was extracted with ethyl acetate (EA), dried with anhydrous sodium sulfate (Na_2_SO_4_) and concentrated to obtain the crude product, which was purified by column chromatography (PE/EA = 20/1) to obtain **B**, a light yellow solid, with a yield of 94%, and m.p. 91–92 °C. ^1^H NMR (400 MHz, CDCl_3_) δ 7.71 (s, 1H, benzene H), 7.53–7.40 (m, 4H, benzene H), 7.39–7.31 (m, 1H, benzene H), 5.34 (s, 2H, NH_2_), 4.25 (q, *J* = 7.1 Hz, 2H, CH_2_CH_3_), 1.32 (t, *J* = 7.1 Hz, 3H, CH_2_CH_3_); ^13^C NMR (101 MHz, CDCl_3_) δ 164.41, 148.92, 140.47, 137.43, 129.57, 127.93, 123.61, 95.95, 59.50, 14.38; HRMS (ESI): *m*/*z* calcd for C_12_H_14_O_2_N_3_ [M+H]^+^, 232.10805; found, 232.10805.

#### 4.3.2. General Procedure for the Preparation of Intermediate **C**

Intermediate **B** (44.5 mmol) and 80% hydrazine hydrate (35 mL) were mixed, reacted at 125 °C for 3 h and then cooled and filtered. The filter cake was washed with ethanol (40 mL) and dried to obtain intermediate **C**, a white solid, with a yield of 80%, m.p.182–183 °C. ^1^H NMR (400 MHz, DMSO-*d*_6_) δ 9.16 (s, 1H, NHNH_2_), 7.89 (s, 1H, pyrazole H), 7.57–7.50 (m, 4H, benzene H), 7.40 (t, *J* = 16.0 Hz, 1H, benzene H), 6.33 (s, 2H, NHNH_2_), 4.27 (s, 2H, NH_2_); ^13^C NMR (101 MHz, DMSO-*d*_6_) δ 164.88, 149.36, 138.65, 138.42, 129.85, 127.53, 123.52, 96.55; HRMS (ESI): *m*/*z* calcd for C_12_H_12_ON_5_ [M+H]^+^, 218.10364; found, 218.10362.

#### 4.3.3. General Procedure for the Preparation of Intermediate **D**

Intermediate **C** (13.8 mmol) was added to a solution of KOH (1.2 g, 20.7 mmol) in 50 mL ethanol, and CS_2_ (2.5 mL, 41.4 mmol) was slowly added dropwise and stirred for 4 h to obtain crude product, which was added slowly to 30 mL concentrated H_2_SO_4_ and stirred for 7 h at 0 °C. The mixture was filtered and dried to obtain intermediate **D**, a yellow solid, with a yield of 50%, m.p. 210–211 °C. ^1^H NMR (400 MHz, DMSO-*d*_6_) δ 14.44 (s, 1H, SH), 7.80 (s, 1H, pyrazole H), 7.56 (d, *J* = 4.0 Hz, 4H, benzene H), 7.46–7.41 (m, 1H, benzene H), 6.21 (s, 2H, NH_2_); ^13^C NMR (101 MHz, DMSO-*d*_6_) δ 185.01, 155.69, 145.52, 139.45, 138.15, 130.00, 128.20, 124.13, 94.74; HRMS (ESI): *m*/*z* cald for C_9_H_14_N_5_S_2_ [M+H]^+^, 256.06851; found, 256.06848.

#### 4.3.4. General Procedure for the Preparation of Title Compounds **E_1_**–**E_28_**

Intermediate **D** (1.1 mmol), potassium hydroxide (1.6 mmol), H_2_O (5 mL) and acetonitrile (5 mL) were stirred at room temperature (r.t.). Then, halogenated hydrocarbon (1.6 mmol) was added and stirred for 5 h. The mixture was extracted with DCM, washed with saturated brine and dried with Na_2_SO_4_. Then, the crude product was purified by column chromatography (DCM) to obtain title compounds **E_1_**–**E_28_**. The spectral data of compounds **E_1_**–**E_28_** are included in the Appendix A.

### 4.4. In Vivo Antiviral Bioassay against TMV

The curative, protective and inactive activities of title compounds **E_1_**–**E_28_** against TMV in the *N. glutinosa* at a concentration of 500 μg/mL were tested by the half-leaf spot method as described in previous literature [42]. The commercialized antiviral agent ningnanmycin was used as a positive control. Each bioassay was repeated three times, and the results are presented as the means ± standard errors. The detailed procedures are provided in the Appendix A.

### 4.5. Verification of the Protective Activity of ***E_2_*** on N. benthamiana with TMV-GFP

To further investigate the effect of compound **E_2_** on tobacco plants infected with TMV, TMV with a green fluorescent protein (GFP) gene tag was used for the experiment. A solution of **E_2_** (250 μg/mL) was spread on the leaves every 12 h a total of three times. The TMV-GFP agrobacteria (OD_595_ = 0.5–0.8) were injected into the lower second leaf of treated plants after 24 h with a needleless syringe. Then, the cells were observed continuously for 10 days, and photographs of the progress of the viral infection were recorded. The blank control was treated with the same concentration of DMSO. All experiments included three replicates.

### 4.6. N. benthamiana with Different Treatments for a Mechanism Study

*N. benthamiana* plants with normal and healthy growth from 5 to 6 leaves were used for the mechanism study, and all experiments consisted of four groups: the CK group, which consisted of healthy plants treated with PBS solution with the same concentration of DMSO; the T group, which consisted of plants treated with **E_2_** (250 μg/mL) three times (12 h/per treatment); the S group, which consisted of plants infected with TMV (6 × 10^−3^ mg/mL) on the second lower leaf; and the TS group, which consisted of plants first treated with **E_2_** (250 μg/mL) on the whole plant three times (12 h/per treatment) and then infected with TMV. The leaves were sampled from the sixth leaves of treated plants at 9:00–11:30 a.m. at 2, 4 and 6 days post-inoculation (dpi), immediately frozen in liquid nitrogen and stored at −80 °C until use. Measurements were performed in triplicate.

### 4.7. Plant Tissue Morphological Observation

Leaf paraffin section analysis was performed as described by Hou et al. [43]. The leaves were collected from the sixth leaf of *N. benthamiana* at 6 dpi and fixed in stationary liquid (*v*/*v*/*v* = 8/1/1, 50% ethanol/formal/glacial acetic acid) for 48 h. Samples were washed with water, dehydrated with 85, 95 and 100% ethanol and dimethylbenzene and embedded in paraffin. Then, the paraffin samples were sectioned by a Reichert Hisrostat Slicer, and the sections were dewaxed with dimethylbenzene and ethanol solution, stained with 1% (*w*/*v*) safranin T (Macklin Biochemical Co., Ltd., Shanghai, China) and 1% (*w*/*v*) fast green FCF (Solarbio Sci & Tech Co., Ltd., Beijing, China) and photographed with a microscope.

### 4.8. Physiological Parameter Measurement and Stomatal Observation for the Host

A portable photosynthesis system (HED-GH20 photosynthesis system) was used to measure the physiological parameters, including the net photosynthesis (*Pn*), stomatal conductance (*gs*), transpiration rate (*Tr*) and intercellular carbon dioxide concentration (*Ci*), of *N. benthamiana* leaves. The rate of CO_2_ assimilation in the chamber was measured at 9:00–11:30 a.m. on 2, 4 and 6 dpi with a leaf temperature of 28 °C, 75% humidity, PPFD of 500 μmol m^−2^ s^−1^, flow lux of 800 mL min^−1^ and an ambient CO_2_ concentration of 370 µL L^−1^. At 4 dpi, the samples were observed directly under the microscope, and the number of stomata in the subcutaneous epidermis of *N. benthamiana* was counted in 20 visual fields. All experiments were performed in triplicate.

### 4.9. Determination of Chlorophyll, H_2_O_2_ and Malondialdehyde (MDA) Contents

Fresh *N. benthamiana* (0.1 g) was weighed, and the chlorophyll, H_2_O_2_ and MDA contents were measured as described in the chlorophyll kit (Suzhou Comin Bioengineering Institute, Suzhou, China), H_2_O_2_ and MDA kits (Solarbio Sci & Tech Co., Ltd., Beijing, China), respectively. Each experiment was performed in triplicate.

### 4.10. Statistical Analysis

SPSS 18.0, ImageJ 1.49 and Origin 9.0 were used for statistical analysis and graph construction. The error bars represent the standard error of the mean. The physiological index data were statistically analyzed using ANOVA, and significant differences at *p* < 0.05 were determined using Duncan’s multiple-range test. All the results are presented as the mean values ± standard errors (S.E.) of at least three independent experiments.

## 5. Conclusions

In conclusion, a series of novel 1-phenyl-4-(1,3,4-thiadiazole-5-thioether)-1*H*-pyrazole-5-amine derivatives were designed and synthesized. In vivo antiviral activity evaluation revealed that some compounds exhibited excellent protective activity against TMV in tobacco plants, and the EC_50_ value of **E_2_** was 203.5 μg/mL, which was better than that of ningnanmycin (261.4 μg/mL). The mechanism study of the active compounds demonstrated that compound **E_2_** could induce the tight arrangement and alignment of spongy mesophyll and palisade cells, prompting them to contain more chlorophyll, thus enhancing photosynthesis efficiency in TMV-infected tobacco leaves by maintaining stable chlorophyll content and protecting the host plants against viral infection. In addition, compound **E_2_** could induce stomatal closure of the leaves to prevent further viral infection through stomata during TMV infection and effectively reduce the content of peroxides in the infected plants, reducing the damage to the plants caused by oxidation. These active compounds could be used as potential antiviral agents for protecting host plants, and an in-depth mechanism study is ongoing.

## Data Availability

The data that support the findings of this study are available from the corresponding authors upon reasonable request.

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
