# Peer review of "Novel 1,3,4-Thiadiazole Derivatives: Synthesis, Antiviral Bioassay and Regulation the Photosynthetic Pathway of Tobacco against TMV Infection"

_ijms, 2023, doi:10.3390/ijms24108881_

Round 1

Reviewer 1 Report

Some aspects should be clarified:

• clearer presentation of the conditions for the appearance of the virus and the factors that facilitate its appearance.

• a clear presentation of the mechanism of action of the virus in the destruction of tobacco crops.

• if the factors that determine the quality of the soil, as well as the factors that condition the tobacco culture can intervene in blocking the action of the virus on the tobacco plants?

Author Response

Thank you very much for providing further suggestions on our manuscript. The paper has been modified carefully following your comments. Each comment has been taken care of and answered by referring to the relevant part of the manuscript. All the changes were based on the original document and written in red in revised manuscript.

Thank you for your kind help throughout the evaluation process of our manuscript.

Reviewer 1:

Comments and Suggestions for Authors:

  1. Clearer presentation of the conditions for the appearance of the virus and the factors that facilitate its appearance.

Response: Thank you very much for your suggestion. The tobacco mosaic virus was purchased from Wuhan Institute of Virology, Chinese Academy of Sciences and preserved in N. tabacum cv. K326. The pure virus extracted from the leaves was used directly in the experiment and stored in a -80 oC refrigerator. In addition, humidity and temperature have the greatest influence on virus infection, and the temperature mainly affects the speed of virus invasion. The virus invades quickly and has a strong invasion rate in a suitable temperature range, so the host plant is kept in a climate with 85% humidity and a temperature of 28/25 oC day and night, which ensures both plant growth and virus infection.

  1. A clear presentation of the mechanism of action of the virus in the destruction of tobacco crops.

Response: Thank you very much for your suggestion. After inoculation by friction, the virus enters the host directly through the wounds and stomata of the leaf and continues to multiply and spread to infect the entire host. Infected plants exhibit curling, yellowing, and diseased leaf spots, resulting in weakened photosynthesis and stunted development that eventually withers.

  1. If the factors that determine the quality of the soil, as well as the factors that condition the tobacco culture can intervene in blocking the action of the virus on the tobacco plants?

Response: Thank you very much for your suggestion. Three kinds of tobacco plants (Nicotiana glutinosa, Nicotiana benthamiana, and Nicotiana tabacum cv. K326) used in experiments were cultivated and maintained in growth chambers at a temperature of 28 oC, a day period with a 16 h photoperiod, a night cycle of 8 h, 25 oC and 85% humidity. In addition, the soil used for the experiments did not interfere with the virus, and the experiments were performed under the same suitable conditions for plant growth, thus excluding the influence of climatic factors on the viral infection, which is hindered by the active compound.

Reviewer 2 Report

REVIEWER'S REPORT

Manucsript title: Novel 1,3,4-thiadiazole Derivatives: Synthesis, Antiviral Bioas
say, and Regulation the Photosynthetic Pathway of Tobacco against TMV Infection (Authors: Huanlin Zheng, Fanglin Wen , Chengzhi Zhang, Rui Luo, Zhibing Wu and *???)

  In this manuscript, a set of novel 1-phenyl-4-(1,3,4-thiadiazole-5-thioether)-1H-pyrazole-5-amine derivatives were synthesized and their  protective activity against  tabacco mosaic virus (TMV) was examined. The protective activity of some compounds were defined to be higher than that of commercially agent ningnanmycin.  This manuscript, in my opinion, could be accepted with a few corrections and supplementations.

 One of the coauthors' names appears to be missing.

    In Materials and Methods, the methodology for defining the antiviral activity of the title compounds, in my opinion, is given in a formal manner.  It is unclear how the EC50 values were determined. Typically, EC50 is defined through data approximation with logistic equations, providing CL50 values (at activity inflection points against varying concentrations of the compounds).  For representation, one or more Figures showing the activity dependence upon varying concentrations of the most active compound(s) along with the approximation curves should be represented.

   It is not thoroughly clear about linear regression equations provided in Tables 2 and 3. Do these equations describe the defined activity dependences of the compounds on their varied concentrations, which are used to calculate CL50 values? 

Albeit the English is intelligible and proper in my opinion, I recommend that the entire text of the article be carefully proofread.

Author Response

Thank you very much for providing further suggestions on our manuscript. The paper has been modified carefully following your comments. Each comment has been taken care of and answered by referring to the relevant part of the manuscript. All the changes were based on the original document and written in red in revised manuscript.

Thank you for your kind help throughout the evaluation process of our manuscript.

Reviewer 2:

Comments and Suggestions for Authors:

Manuscript title: Novel 1,3,4-thiadiazole Derivatives: Synthesis, Antiviral Bioassay, and Regulation the Photosynthetic Pathway of Tobacco against TMV Infection (Authors: Huanlin Zheng, Fanglin Wen, Chengzhi Zhang, Rui Luo, Zhibing Wu and *???)

In this manuscript, a set of novel 1-phenyl-4-(1,3,4-thiadiazole-5-thioether)-1H-pyrazole-5-amine derivatives were synthesized and their protective activity against tabacco mosaic virus (TMV) was examined. The protective activity of some compounds were defined to be higher than that of commercially agent ningnanmycin. This manuscript, in my opinion, could be accepted with a few corrections and supplementations.

  1. One of the coauthors' names appears to be missing.

Response: Thank you very much for your suggestion. There was an error, and no co-author’s name was left out.

  1. In Materials and Methods, the methodology for defining the antiviral activity of the title compounds, in my opinion, is given in a formal manner. It is unclear how the EC50 values were determined. Typically, EC50 is defined through data approximation with logistic equations, providing CL50 values (at activity inflection points against varying concentrations of the compounds). For representation, one or more Figures showing the activity dependence upon varying concentrations of the most active compound(s) along with the approximation curves should be represented.

Response: Thank you very much for your suggestion. The methodology used to define the antiviral activity of the title compounds was showed in section “1. Detailed Antiviral Biological Assay Methods” of the supporting information. The detailed method of EC50 values determination was added in section “1.4 Determination of EC50 Values” of the supporting information as follows. As an example, the EC50approximation curve of target compound E2 for its protective activity against TMV is shown in Figure 1 below.

1.4 Determination of EC50 Values

Taking the process of EC50 value testing for the target E2 as an example, 4 mg compound E2 was accurately weighed and dissolved in 80 µL of DMSO to give a mother liquor. Then, 40 µL solution was taken out and mixed with 4 mL 0.1% TW-80 water to give the first concentration (500 µg/mL); after adding 40 µL DMSO to the remaining 40 µL of solution, 40 µL solution was taken out and mixed with another 4 mL 0.1% TW-80 water to give the second concentration (250 µg/mL). According to the same operation to prepare solutions of different concentrations. And the EC50 values of target compounds against TMV were measured. There are three replicates for each compound.

Figure 1. The EC50 approximation curves of target compound E2.

  1. It is not thoroughly clear about linear regression equations provided in Tables 2 and 3. Do these equations describe the defined activity dependences of the compounds on their varied concentrations, which are used to calculate CL50 values?

Response: Thank you very much for your suggestion. The EC50 values were calculated from linear equations determining the inhibitory activity of the antiviral compounds at five concentration gradients, respectively, and the inhibitory activity at different concentrations were shown in Tables S1 and S2 in the supporting information as follow:

Table S1. The EC50 values of some title compounds for curative activity a (μg/mL).

* ningnanmycin; a Each experiment was performed in triplicate.

Table S2. The EC50 values of some title compounds for protective activity a (μg/mL).

* ningnanmycin; a Each experiment was performed in triplicate.

Comments on the Quality of English Language:

  1. Albeit the English is intelligible and proper in my opinion, I recommend that the entire text of the article be carefully proofread.

Response: Thank you very much for your suggestion. The manuscript has been polished by AJE (Wiley Language Editing Services). The verification code is 2898-6409-DDDB-6FA1-35EA. The editing certificate are provided as follow:

Other corrections made by the authors:

  1. Supporting information, line 28, “500 µg/mL” was corrected to “1.0 mg/mL”; line 67, “2. Crystallographic Data of Title Compound E3” was changed to “3. Crystallographic Data of Title Compound E3”; line 68, “Table S2” was changed to “Table S3”; line 69, “3. Protective and Curative Activity Images of Title Compounds” was changed to “4. Protective and Curative Activity Images of Title Compounds”; line 76, “4. 1H NMR, 13C NMR and HRMS Data of Intermediates and Title Compounds” was changed to “5. 1H NMR, 13C NMR and HRMS Data of Intermediates and Title Compounds”; line 257, “5. Copies of Intermediates and Title Compounds” was changed to “6. Copies of Intermediates and Title Compounds”.
